# Growth Stimulation, Phosphate Resolution, and Resistance to Fungal Pathogens of Some Endogenous Fungal Strains in the Rhizospheres of Medicinal Plants in Vietnam

**DOI:** 10.3390/molecules27165051

**Published:** 2022-08-09

**Authors:** Nguyen Thi Mai Huong, Pham Thi Thu Hoai, Phan Thi Hong Thao, Tran Thi Huong, Vu Duc Chinh

**Affiliations:** 1University of Economics—Technology for Industries (UNETI), Hanoi 11622, Vietnam; 2Institute of Biotechnology, Vietnam Academy of Science and Technology, 18 Hoang Quoc Viet Road, Cau Giay District, Hanoi 11307, Vietnam; 3Institute of Materials Science, Vietnam Academy of Science and Technology, 18 Hoang Quoc Viet Road, Cau Giay District, Hanoi 11307, Vietnam; 4School of Materials Science and Energy Engineering, Graduate University of Science and Technology, Vietnam Academy of Science and Technology, 18 Hoang Quoc Viet Road, Cau Giay District, Hanoi 11307, Vietnam

**Keywords:** endophytic fungi, medicinal plants, phosphate resolution, IAA, *Penicillium* sp., *Talaromyces* sp., *Trichoderma* sp.

## Abstract

Endophytic fungi are recognized for their many potential applications in agriculture, such as supporting cropland expansion and increasing the yield and resistance of plants by creating antibiotics that inhibit the growth of pathogenic microorganisms. In addition, they can produce enzymes that break down hard-to-solubilize substances within soil, dissolve phosphates, fix nitrogen, reduce metals, and produce hormones that promote plant growth (auxin, cytokinin, and gibberellins) to keep crops healthy. In this report, three strains of endophytic fungi, namely, N1, N2, and N3, were isolated from the roots of *Stevia rebaudiana (Bert.) Hemsl.*, *Polyscias fruticosa*, and *Angelica dahurica* in some localities in Vietnam. Through a screening process, it was found that they can produce high levels of indole acetic acid (IAA), resolve phosphates, and resist disease, and they were selected to as an alternative to chemical fertilizers to make probiotics in order to increase medicinal plant yields. The results show that the three strains of fungi have the ability to degrade phosphate to 341.90, 1498.46, and 390.79 ppm; the content of IAA produced in the culture medium reached 49.00, 52.35, and 33.34 ppm. Based on some morphological characteristics and an internal transcribed spacer gene sequence analysis of the fungal strains, N1, N2, and N3 were named *Penicillium simplicissimum CN7*, *Talaromyces flavus BC1*, and *Trichoderma konilangbra DL3*, respectively, which have the ability to inhibit the growth of pathogenic fungal strains, such as fungus *C. gloeosporioides (CD1)*, fungus *F. oxysporum*, fungus *L. theobromae* N13, and *N. dimidiatum*. They grow significantly over a period of 5 to 6 days.

## 1. Introduction

The demand for medicinal herbs and medicinal products is increasing as the world attempts to reshape research methods in order to find new drugs rather than focusing solely on the synthesis of chemicals in the laboratory, a procedure with many difficulties resulting in toxic chemicals. Research and experiments on nature conducted to identify new bioactive substances with more favorable, less toxic properties are receiving a lot of attention.

Endophytic bacteria colonize plants and live inside them for part of or throughout their life without causing any harm or disease to their hosts. Endophytes promote plant growth and fitness through the production of phytohormones or biofertilizers, or by alleviating abiotic and biotic stress tolerance. Endophytic bacteria can be used for the phytoremediation of environmental pollutants or for the control of fungal diseases by the production of lytic enzymes, such as chitinases and cellulases, and their wide host range allows for a broad spectrum of applications to agriculturally and pharmaceutically interesting plant species. More recently, endophytic bacteria have also been used to produce nanoparticles for medical and industrial applications [1]. Endophytic bacteria can build a symbiotic association with their host to improve host-plant salt tolerance. These endophytic strains exhibit plant-growth-promoting activities, including phosphate solubilization, ammonia production, phytopathogen biocontrol, extracellular enzymatic activities, and indole-3-acetic acid production, under normal and salinity stress conditions [2].

Vietnam has advantages with its nature, natural conditions, soil, and climate, with more than 5000 types of plants used in disease prevention and healing. Vietnam is a strong player in the world medicinal plant industry [3]. Therefore, increasing the productivity and yield of medicinal plants is an issue of concern and direction. Endophytes can generate a cornucopia of marvelous bioactive secondary metabolites useful for humankind, but their biodiversity and associations with host plants are still elusive. In a study conducted by Tran H.M. et al., culturable endophytic microorganisms associated with 14 medicinal plants that are of high socio-economic value and/or reportedly endemic to northern Vietnam were investigated. Specifically, they isolated endophytic microorganisms by applying surface sterilization methods and identified them based on morphological and rDNA sequence analyses. Agglomerative hierarchical clustering and principal component analysis were used to analyze the correlations between the taxonomic affiliations of the culturable endophytes and the characteristics of their hosts. Most of the culturable endophytes obtained were bacteria (80), and few of those were actinomycetes (15) and fungi (8). Many of them were reported to be endophytes of medicinal plants for the first time. A number of plants (5) were also reported for the first time to contain microbial endophytes, while some plants with powerful pharmaceutical potential were reported to harbor unique endophytes. Furthermore, their results reveal a strikingly close relationship between the compositions of bacterial and fungal isolates from plants with anti-bacterial activity and those from plants with anti-inflammatory activity, or between the compositions of the microbial endophytic isolates from plants with anti-cancer activity and those from plants with antioxidant activity. Altogether, the results provide new findings that are inspiring for further in-depth studies to explore and exploit the relationships between medicinal plants and their associated endophytes in northern Vietnam and worldwide [4].

One of the solutions to increase the productivity and yield of medicinal plants in particular and plants in general requires the use of products containing capable microorganisms. These microorganisms can fix nitrogen; resolve insoluble phosphates; and synthesize hormones for plant growth, such as indole acetic acid (IAA). However, the use of probiotics also has many advantages, such as low cost, easy use, high effects, and good safety, as well as being a replacement for chemical fertilizers.

Promoting plant growth with the use of microorganisms is considered a potential tool for sustainable agricultural production and development trends for the future. Experiments have been conducted to learn more about microbial adaptations to promote schizophrenia; root mechanisms; and physiological, biochemical, and growth-stimulating effects in plants. They are used to produce microbial fertilizers for sustainable agriculture and horticulture production. Among the microorganisms associated with plants, the relationship between fungi and plants is extremely common and attracts much research attention due to the ability of fungi to solubilize nutrients in soil, promote plant growth, act as biological control agents, and activate plant systemic resistances to biotics [5,6,7,8]. Endophytic fungi have demonstrated their role in increasing agricultural crop yields. They can colonize the intercellular and intracellular spaces of plants; alter the metabolic mechanism of host plants; promote nutrient acquisition; promote the assimilation of phosphorus, as well as other ions, such as zinc, copper, and nitrogen; protect plants from fungal diseases and nematodes; improve drought tolerance; enhance growth; and promote nutrient acquisition [6,9,10,11,12]. Endophytic fungi also manifest themselves as plant stimulants to produce self-protective active substances commonly known as phytoalexins [13].

Phosphorus is the second most important macronutrient after nitrogen for plants. It is an indispensable part of plants in general and of medicinal plants in particular. However, the main source of phosphorus is found in soil in the form of complexes with iron, calcium, and aluminum, which cannot be used by plants [14]. Deficiency phosphorus can slow plant growth and reduce leaf biomass [15,16]. Using chemical fertilizers to supply plants with their nutritional needs is extremely popular in agriculture, but it is not environmentally friendly due to non-biodegradable inorganic compounds. Using fungal strains is seen as a useful alternative solution due to their ability to decompose phosphates by creating various organic acids, such as tartaric acid, succinic acid, oxalic acid, malic acid, 2-ketogluconic acid, glyoxylic acid, gluconic acid, fumaric acid, citric acid, and alpha-ketobutyric acid, and the enzyme phosphatase. They are phosphate soluble, and plants can use the dissolved inorganic phosphate form.

IAA is the most common auxin-class phytohormone, and it plays an important role in the growth of plants, as well as contributing to xylem and phloem structures. IAA is a product of L-tryptophan metabolism, it is produced by some microorganisms around the roots, supporting plant growth. It promotes root elongation by increasing the number of lateral roots that take up nutrients [17].

Among 27 fungal strains isolated from the root and rhizospheres of medicinal plants (data not shown here), three strains, namely, N1, N2, and N3, which have the highest ability to produce IAA growth stimulants and degrade insoluble phosphate, were selected. Their biggest advantages are their ability to produce IAA, produce its growth stimulants, and decompose insoluble phosphates.

The selected fungal strains are resistant to pathogenic fungi, degrade phosphate, and produce growth stimulant IAA from the roots of very promising medicinal plants for application in the production of plant support preparations [14,18]. Furthermore, research into the mycorrhizal zone on medicinal plants is essential in identifying diversity and finding potential strains for the production of products with the aim of improving the productivity and quality of medicinal plants, preserving the precious medicinal resources in Vietnam.

## 2. Results and Discussion

### 2.1. Identification of Most Potent Endophytic Fungal Strains

Endophytic microorganisms contribute to the growth of plants and the recovery of plant health by using different strategies, including controlling phytopathogens and secreting phytohormones, such as IAA, cytokinins, and gibberellic acids. Additionally, endophytes can reinforce plant growth by nitrogen fixation, phosphate solubilization, nutrient cycling, and novel and bioactive metabolite secretion. Secondary metabolites secreted by endophytic microbes have various biotechnological applications [19,20,21].

We isolated the rhizomic symbiotic fungal systems of three plant samples and found 27 specific fungal strains: 07 strains of *Stevia rebaudiana* (rhizospheres: 4 strains; soil: 3 strains), 11 strains of *Polyscias fruticosa* (roots: 6 strains; soil: 5 strains); and *Angelica dahurica*: 9 strains (roots: 6 strains; soil: 03 strains). Felde et al. [22] demonstrated that the addition of endophytic fungi *Trichoderma atroviride* and *F. oxysporum* improved and increased banana yield, and it reduced the number of parasitic nematodes on banana *Radopholus similis*. The endophytic fungi *Sebacina vermifera*, *Piriformospora indica*, *Colletotrichum*, and *Penicillium* have better plant-growth-promoting effects under adverse conditions due to their ability to synthesize enzymes and plant growth promoters, such as *auxins*, *cytokinins*, and *gibberellins* [23]. The fungal strain *Talaromyces flavus* has also been selected as a promoter to promote the growth of cotton and potato plants due to its high biosynthetic capacity of IAA [24]. Thus, we selected the fungal strains N1, N2, and N3, which have the highest ability to produce IAA growth stimulants and to degrade insoluble phosphate. In general, the fungal strains isolated from the roots were more abundant than those isolated from the soil. The fungal strains, after isolation and cleaning, were grown on potato agar to observe the morphological characteristics and the color of the *mycelium*, as well as the pigment produced in the medium, and the fungi were grouped according to the color of the *mycelium*; the results are shown in Table 1. They were then fermented in potato liquid medium. The results are shown in Table 2.

The results show that strain N1 was closest to the species *Penicillium* genus (100% similarity), strain N2 had 100% similarity with the fungal strain of the *Talaromyces* genus, and strain N3 had a high similarity with the fungal strain of *Trichoderma* (100% similarity). The endophytic fungi strains were *Penicillium simplicissimum CN7*, *Talaromyces flavus BC1*, and *Trichoderma konilangbra DL3*, respectively.

Some fungal species, including P. bilaji, *Penicillium* spp. [25], P. oxalicum [26,27], Aspergillus niger, *Penicillium notatum* [28,29], *Aspergillus awamori* [30], *Penicillium bilaii* [31], *Trichosporon beigelii*, *Rhodotrula aurantiaca*, *Cryptococcus luteolus*, *Zygoascus hellenicus*, *P. purpurogenum var. rubrisclerotium*, *Neosartorya fisheri* and *Candida montana* [32], *Talaromyces aurantiacus*, *Aspergillus neoniger* [33], and *Trichoderma* spp. [34], have been reported to solubilize phosphorus. Babu et al. (2015) [35] also found that *Penicillium menonorum* can enhance plant growth and induce IAA.

Patil et al. conducted a study to evaluate the effects of P-solubilizing fungi and phosphorus levels on the growth, yield, and nutrient content of maize [25]. The field experiment was conducted to test the effects of the P-solubilizing fungi *Penicillium bilaji* and *Penicillium* spp. On the availability of applied P-fertilizer in calcareous soil at MARS, University of Agricultural Sciences, Dharwad, during the *rabi/summer* season of 2009–2010. Seed inoculation with phosphorus-solubilizing fungi, in addition to P_2_O_5_ levels, significantly influenced plant height, the number of leaves per plant, dry matter production, cob length, grain weight per cob, 1000 grain weight, grain yield, and tissue nutrient content (N, P, K, Zn, and Fe) at the tasseling of leaves and the harvest of the whole plant, as well as P uptake at harvest. Stover yield was not significantly influenced by the various treatments. Higher growth and a higher yield of maize were achieved when using P-solubilizing fungi treatment and 100% RD P_2_O_5_ application compared to 0 and 50% RD P_2_O_5_. It was concluded that single and dual inoculations along with P-fertilizer had 20–23% higher maize yields than the control.

Phosphate-solubilizing fungi (PSF) have huge potential to enhance the release of phosphorus from fertilizer. Two PSF were isolated and identified as *Penicillium oxalicum* and *Aspergillus niger* in a study conducted by Li et al. [26]. To enhance the fertilizer value of rock phosphate in alkaline soils, the phosphate (P)-solubilizing fungus *Penicillium oxalicum* was isolated from the rhizosphere soil of rock phosphate mine landfills, and it was tested for its efficacy to solubilize rock phosphate (RP) and to promote the growth of wheat and maize plants grown in soil amended with RP. The results showed that *P. oxalicum* effectively solubilized RP in Pikovskaya (PVK) medium and released a high amount of phosphorus [27]. In a previous study, the *Azotobacter* (SR-4) strain was found to be an efficient nitrogen fixer, as 35.08 mg of nitrogen per gram of carbon was produced after 72 h of fermentation. Similarly, the *A. niger* strain was found to excrete extracellular phosphate-solubilizing enzymes, such as phytase (133UI in 48 h of fermentation) and phosphatase (170UI in 48 h of fermentation), which can solubilize rock phosphate and make it available to plants. Furthermore, plants co-inoculated with both N-fixing *Azotobacter* and phosphorus-solubilizing *A. niger* have been found to have enhanced performance compared to those treated with each biofertilizer alone [28]. PSF function in the soil phosphorus cycle by increasing the bioavailability of soil phosphorous for plants. In a previous study, *Aspergillus niger* and *Penicillium notatum* were tested for their efficacy to solubilize tricalcium phosphate (TCP) in vitro, as well as for their ability in vivo to promote the growth of groundnut (*Arachis hypogaea*) plants grown in soil amended with TCP. The results showed a high solubilizing index in agar plates. Moreover, they effectively solubilized TCP in PVK liquid medium and released considerable amounts of P into the medium. A pot experiment showed that the dual inoculation of phosphate-solubilizing fungi (*A. niger and P. notatum*) significantly increased the dry matter and yield of groundnut plants as compared to control soil [29].

In a previous study, a phosphate-solubilizing fungus, *Aspergillus awamori* S29, was isolated from the rhizosphere of *mungbean*. The phosphate-solubilizing activity of *A. awamori* S29 in liquid was 1110 mg/L for TCP. The organism was able to solubilize various inorganic forms of phosphate at a wide range of temperatures. Among the various insoluble phosphate sources tested, di-calcium phosphate was solubilized the most, followed by TCP. *A. awamori* S29 had a significant effect (*p*  <  0.05) on mungbean growth, total P, and plant biomass under pot conditions, although no obvious differences in available P in the soil and the number of leaves were found compared to the control [30]. In view of this, field experiments were conducted to evaluate the effect of seed inoculation with PSF (*Penicillium bilaii*) at different rates of fertilizer P on P content in leaves and the grain yield of irrigated wheat in India. The soil was low in Olsen P at the Bathinda site and had a medium level at the Ludhiana site. In the no-P treatment, PSF significantly increased grain yield by 12.6% over the non-inoculated control. The effect of PSF on grain yield was generally more pronounced in soil with a low Olsen-P level compared to soil with a medium Olsen-P level. The inoculation of PSF along with 50% P fertilizer increased wheat yield equivalent to that of 100% P with no PSF. Spike density was significantly higher in PSF + 50% P than all other treatments. There is a need to study the long-term effect of *Penicillium bilaii* on P-fertilizer saving in wheat on soils varying in P availability, pH, and P fixation capacity for different wheat-based cropping systems [31]. In the study conducted by Zhang et al. (2018), two PSF, TalA-JX04 and AspN-JX16, were isolated from the rhizosphere soil of moso bamboo (*Phyllostachys edulis*), which is widely distributed in P-deficient areas in China, and they were identified as *Talaromyces aurantiacus* and *Aspergillus neoniger*. The two PSF were cultured in potato dextrose liquid medium with six types of initial pH values ranging from 6.5 to 1.5 to assess acid resistance. Both PSF were incubated in PVK liquid media, with different pH values, containing five recalcitrant P sources, namely, Ca_3_(PO_4_)_2_, FePO_4_, CaHPO_4_, AlPO_4,_, and C_6_H_6_Ca_6_O_24_P_6_, to estimate their P-solubilizing capacity. The P-solubilizing capacity of TalA-JX04 was the highest in the medium containing CaHPO_4_, followed by Ca_3_(PO_4_)_2_, FePO_4_, C_6_H_6_Ca_6_O_24_P_6_, and AlPO_4_, in the five types of initial pH treatments, while the recalcitrant P-solubilizing capacity of AspN-JX16 varied with initial pH. Meanwhile, the P-solubilizing capacity of AspN-JX16 was much higher than that of TalA-JX04. The pH of the fermentation broth was negatively correlated with P-solubilizing capacity (*p *< 0.01), suggesting that the fungi promote the dissolution of P sources by secreting organic acids. Their results showed that TalA-JX04 and AspN-JX16 can survive in acidic environments and that both fungi had a considerable ability to release soluble P by decomposing recalcitrant P-bearing compounds [33]. Acidic soils rapidly retain applied phosphorus fertilizers and, consequently, present a low availability of this nutrient to plants. The use of phosphate-solubilizing microorganisms to aid plant phosphorus absorption is a promising sustainable strategy for the management of P deficiencies in agricultural soils. In a previous study, 19.5% of isolated *Trichoderma* strains were able to solubilize phosphate. In addition, those strains produced different organic acids during the solubilization process. *Trichoderma* spp. strains showed positive responses in the promotion of soybean growth—from 2.1% to 41.1%—as well as in the efficiency of P uptake, up to 141% [34].

Thus, it can be seen that the fungal strains isolated from the roots of medicinal plants in Vietnam belong to the same species as the fungal strains previously isolated and proven to be able to degrade insoluble phosphate and produce IAA across the world. A phylogenetic tree was built using MEGA X software (Figure 1).

### 2.2. Growth Dynamics of Fungal Strains

Fungal strains often have different optimal biomass growth times. Determining the optimal growth time has a great influence on the practical application of biomass for inoculant production. Therefore, three strains, *Penicillium simplicissimum CN7, Talaromyces flavus BC1*, and *Trichoderma konilangbra DL3*, were studied for their growth dynamics.

Based on the results, it was found that all three strains of fungi produced the highest biomasses at 144 h after the culture on PDB medium and shaking at 200 rpm at 30 °C, with the biomasses reaching 175.4 g/L for *Penicillium simplicissimum CN7*, 108.4 g/L for *Talaromyces flavus BC1*, and 91.2 g/L for *Trichoderma konilangbra DL3*. This result provides a significant contribution by demonstrating how to obtain strains in production and use them in practice (Figure 2). In this study, after 168 and 192 h, the fermentation pH tended to increase gradually, and the strains *Penicillium simplicissimum CN7* and *Trichoderma konilangbra DL3* formed pellets with small, smooth, tightly coiled spheres, which obstructed the respiratory process and affected the biological production growth of the fungal strains, while the filamentous *Talaromyces flavus BC1* strain was less affected.

In a previous study conducted by Zhang et al., 2018, the fungus strain *Talaromyces* JX04, which was found to degrade insoluble phosphates isolated from the soil around bamboo roots in Jiangxi, China, produced the highest biomass of 22.88 gL^−1^ on PDA medium at 25 °C after 6 days [33]. This biomass was lower than that produced by the fungus *Talaromyces flavus BC1* in the same culture conditions. However, the pH and time affected the biomass production when culturing in PDB medium at 25 °C for 6 days, according to the reduction trend.

The fungal strains *Penicillium* sp. and *Trichoderma* sp. also have the ability to produce cellulases [18,36,37,38,39], pectinase enzymes [40],and xylanase enzymes [41,42], which are all plant growth promoters.

### 2.3. Filamentous Propagation of Fungal Strains

To evaluate growth ability, *Penicillium simplicissimum CN7, Talaromyces flavus BC1*, and *Trichoderma konilangbra DL3* fungal strains were cultured on PDA medium plates at 25 °C for 7 days.

On the PDA medium plates, the *Penicillium simplicissimum CN7* strain grew into a thin hyphal cell wall, with short and cottony mycelium. The *Penicillium simplicissimum CN7* strain was initially white and became green-brown. Then, it changed to a gray-brown after 7 days of culture. The *Penicillium simplicissimum CN7* strain had a rapid growth rate, reaching 34.4 µm/h (Figure 3 and Table 3).

Similarly, *Talaromyces flavus BC1* rapidly grew into the thin hyphal cell wall, with long, straight, and cottony mycelium. Initially, the *Talaromyces flavus BC1* strain was white, and then it became yellow after 10 days. The growth rate of the *Talaromyces flavus BC1* colonies on agar plates reached 43.88 µm/h (Figure 3 and Table 3).

The *Trichoderma konilangbra DL3* strain formed a thin hyphal cell wall with short mycelium on the PDA medium. The mycelium of the strain *Trichoderma konilangbra DL3* was initially white and became bright green. *Trichoderma konilangbra DL3* had a fast growth rate of 106.02 µm/h (Figure 3 and Table 3).

### 2.4. Antagonistic Activity against Pathogenic Fungi

An evaluation of the ability to antagonize pathogenic fungal strains was carried out to determine the effect of inoculants in agriculture. Endophytic fungi and pathogenic fungi were co-cultured on PDA plates to investigate their antagonistic abilities. Antagonistic fungal strains inhibited the growth of pathogenic fungi (Figure 4).

The results show that the fungus *Trichoderma konilangbra DL3* could inhibit the fungus *C. gloeosporioides* (CD1), which causes anthracnose on fruit; well antagonize the fungus *F. oxysporum*, which causes wilt disease; well inhibit the fungus *L. theobromae N13*, which causes stem rot disease on mango; and weakly inhibit *N. dimidiatum* fungus, which causes brown spot disease on dragon fruit. *Talaromyces flavus BC1* could inhibit *C. gloeosporioides* and is resistant to *L. theobromae N13**. Penicillium simplicissimum CN7* could inhibit *L. theobromae N13* fungus, and it had weakly inhibitory or non-inhibitory effects on other fungi.

Gajera et al. (2016) [43] reported that *Trichoderma viride* had an antioxidant response to *Aspergillus niger*, which causes collar rot in groundnut. In another study, *Trichoderma viride* was reported to inhibit the growth of the mycelium of *Sclerotium rolfsii, Fusarium solani*, *and Rhizoctonia solani*. In addition, the alcohol extract of the mycelium showed strong antibacterial activity against *Bacillus subtilis*, *E. coli*, and *Pseudomonas fluorescens*, and these substances also showed significant antifungal activity against *Candida albicans, Fusarium oxysporum, Rhizoctonia solani*, *and Fusarium solani*. Furthermore, Ribeiro et al. (2018) [44] found that *Diaporthe* sp. and *Curvularia* sp. inhibited Gram-negative bacteria that cause disease with Gram-positive bacteria, such as *Staphylococcus epidermidis*, *Pseudomonas aeruginosa, Staphylococcus aureus*, and *E. coli*. Iswarya and Ramesh (2019) reported that *Penicillium* dalea EF4 isolated from the seaweed *Enteromorpha flexuosa* Linn showed antibacterial activity against *Staphylococcus aureus* and *Pseudomonas aeruginosa* [45].

In a study conducted Ramalingam et al. (2014) [46], *Penicillium* spp. isolated from *legume* roots could produce amino acids. Strains RDA01, NICS01, and DFC01 fostered the growth of cycad. Furthermore, NICS01 minimized salinity and protected sesame plants from *Fusarium* infection. The results of this study suggest that NICS01 can be used as a biofertilizer and a biocontrol agent, as it enhances plant growth under biotic and abiotic stress conditions. Therefore, the application of NICS01 is extremely favorable for crop cultivation, even in saline environments.

In addition, the results of previous studies have shown that *Penicillium* spp. can prevent *Fusarium* infections in crops [47,48]. An evaluation of the feasibility of secreting multiple antibiotics, such as 2,4-diacetylphloroglucinol, pyoluteorin, pyrrolnitrin, pyocyanin, oligomycin, and phenazine, by using biological control agents that can inhibit pathogenic fungi was carried out [49].

In another study conducted by Farias et al. [50], plant-growth-stimulating fungi, such as *Trichoderma Asperella*, *Pochonia chlamydosporia, Purpureocillium lilacinum, Metarhizium anisopliae,* and *Beauveria bassiana*, were applied to maize, sugarcane, soybean, and tomato crops, which were grown under two conditions: with and without a fungi complex.

Using fungi as biocontrol agents is extremely beneficial due to their diversity and metabolic efficiency, and it enhances the chances of finding isolates for environmentally friendly biological control because they are mainly decomposers [51]. The fungi of the genera *Aspergillus, Fusarium, Gliocladium, Petriella*, and *Trichoderma* are known to be important biocontrol agents [52]. The biocontrol activity of *Verticillium leptobactrum* against green wilt caused by *Fusarium oxysporum* and *F. lycopersici* was demonstrated by Hajji-Hedfi et al. [53].

Furthermore, many fungal biocontrol agents are also available as commercial products, such as *Verticillium lecanii*, *Trichoderma polysporum*, *Trichoderma gamsii*, *Trichoderma asperellum*, *Purpureocilium lilacinum*, *Phlebiopsis gigantean*, *Paecilomyces lilacinus*, *Metarhizium anisopium*, *anisopliae*, *yeasts*, *Cranberry mushroom*, *Gliocveni pullulans*, and *Ampelomyces quisqualis* [54,55]. The use of fungi as biological control agents provides a safe and environmentally friendly strategy for sustainable agriculture. Furthermore, the potential uses of fungi can be explored to enhance agricultural productivity and metabolite production [56].

## 3. Materials and Methods

### 3.1. Materials

The fungal strain *Penicillium simplicissimum CN7* was isolated from the rhizospheres of *Stevia rebaudiana (Bert.) Hemsl.* (Hung Yen and Vinh Phuc provinces, Vietnam), *Talaromyces flavus BC1* was isolated from the roots of *Polyscias fruticosa* (Nam Dinh and Lao Cai provinces, Vietnam), and *Trichoderma konilangbra DL3* was isolated from the roots of *Angelica dahurica* (Ninh Binh, Phu Tho, Lao Cai provinces, Vietnam). They are phytopathogenic fungi.

### 3.2. Methods

#### 3.2.1. Isolation of Endophytic Fungal Strains

Root isolation: Rinse the sample under running water for 5 min. When the surface of the sample is clean, soak the sample in 70° alcohol for 1 min. Allow the sample to dry in a sterile incubator. Using sterile tools, cut the root samples into small pieces 5–10 mm long, and place them on a Petri dish with an isolation medium; then, transfer them to an incubator at 30 °C for 48–72 h.

Soil sample isolation: Take a small amount of soil, put it into a test tube containing 9 mL of sterile distilled water, and shake well. Pipette a small amount of the solution into a Petri dish with an isolation medium, and then transfer it to an incubator at 30 °C for 48–72 h.

#### 3.2.2. Determination of Phosphate-Solubilizing Activity

Cultures of endophytic fungi were grown on PVK medium (10 g glucose; 5 g Ca_3_(PO_4_)_2_; 0.5 g (NH_4_)_2_SO_4_; 0.2 g KCl; 0.1 g MgSO_4_ 7H_2_O; 0.002 g MnSO_4_; 0.002 g FeSO_4_; 0.5 g yeast extract; pH 6.8–7.0) for 7–10 days at 25 °C in a shaker at 200 rpm. Post-fermentation fluid was centrifuged at 6000 rpm for 20 min to obtain cell-free supernatants. The phosphorus content in the supernatant was determined using a photometric method after the citrate radicals decomposed [57]. The photometric method employs UV/VIS/NIR spectroscopy, a powerful analytical technique used to determine the optical properties (transmittance, reflectance, and absorbance) of liquids and solids. This measurement is used to determine the amount of an analyte in a solution or liquid.

#### 3.2.3. Analysis of Indole Acetic Acid Content

To determine the content of IAA produced by the endophytic fungi, a colorimetric technique was performed with Van Urk Salkowski reagent using Salkowski’s method following Brick et al. (1991) [58]. Endophytic fungi were grown in medium to assess the ability to grow IAA promoters (20 g difco peptone; 1.15 g K_2_HPO_4_; 1.5 g MgSO_4_; 1.5% (*v*/*v*) glycerol; 0.1 g tryptophan), and they were incubated at 200 rpm for 48 h at 30 °C. Post-fermentation fluid was centrifuged at 5000 rpm for 10 min after incubation. The supernatant was used to determine the IAA content in the sample. Then, 1 mL of sample solution was added to 2 mL of Salkowski reagent (2% of 0.5 M FeCl_3_ in 35% HClO_4_ solution) and kept in the dark. The optical density (OD) was recorded at 530 nm after 30 min. The color of the post-reaction mixture changed from light pink to red, depending on the IAA content in the culture solution. For the development of a standard curve, the above procedure was followed with a known concentration of IAA.

#### 3.2.4. Morphological and Growth Characteristics

The fungi were grown on PDA medium plates (5 g potato, 5 g peptone, 20 g glucose; pH 6.5–7.0) at 25 °C. From days 2 to the end of cultivation (days 7), the form, size, and color of the fungal colonies were observed and recorded.

The growth rate of the mycelia was determined according to the method described by Schwantes and Salttler (1971): *V* = Δ*X*/Δ*T*, where *V* is the growth rate of the mycelia (µm/h); Δ*X* is the radius of the colony (µm); and Δ*T* is the cultivation time (hours) [59].

#### 3.2.5. Determination of Antifungal Activity

The antifungal activity of the 3 endophytic fungi against XB1 (*L. thebromae* N13), CD1 (*C. gloeosporioides*), Fu (*F. oxysporum*), and DNTL (*N. dimidiatum*) was determined by modifying the dual-culture plate antagonism. They were cultured on PDA plates at 25 °C for 7 days in the dark. Then, endophytic fungi and pathogenic fungi strains were co-cultured in PDA medium plates by being placed symmetrically on each side of the Petri dish and being incubated at 25 °C. The growth of the fungi strains was observed and recorded regularly every day until the antagonistic fungi strains stopped growing. The experiment was repeated three times and averaged (Zhang et al.) [60].

#### 3.2.6. Molecular Identification

The three fungal strains *Penicillium simplicissimum CN7, Talaromyces flavus BC1*, and *Trichoderma konilangbra DL3* were identified using the molecular biology method. The genomic DNAs of the fungal strains were isolated for PCR amplification, and the amplified PCR products contained bands 800 in length, similar to the size of the internal transcribed spacer gene region. They were compared with the homologous nucleotide sequences on GenBank using the BLAST SEARCH program (https://blast.ncbi.nlm.nih.gov/Blast.cgi) (accessed on 15 July 2021).

The genomic DNAs of the fungi *Penicillium simplicissimum CN7, Talaromyces flavus BC1*, and *Trichoderma konilangbra DL3* were extracted using the alkaline extraction method [61], and they were subjected to PCR to amplify the 5.8S rDNA gene using two primers: BF (5′-CTTGGTCATTTAGAGGAAGTAA-3′) and BR (5′-CAGGAGACTTGTACACGGTCCA-3′) [62]. The thermal cycling program was as follows: initial denaturation at 95 °C for 5 min, followed by 30 cycles; denaturation at 95 °C for 90 s; primer annealing at 55 °C for 90 s; extension at 72 °C for 2 min; and a final extension step of 72 °C for 8 min. The PCR reaction products were examined by electrophoresis in 1% (*w*/*v*) agarose gel, and the bands were stained with ethidium bromide. The PCR products were sequenced by Axil Scientific Pte., Singapore. The sequences of the fungi were compared with similar sequences from GenBank using the BLAST program to identify fungal species. The fungal ITS sequences of *Penicillium simplicissimum CN7*, *Talaromyces flavus BC1*, and *Trichoderma konilangbra DL3* in this study were deposited in GenBank under accession numbers OL795992, OL795993, and OL875113, respectively.

## 4. Conclusions

Endophytic fungi strains *Penicillium simplicissimum CN7, Talaromyces flavus BC1*, and *Trichoderma konilangbra DL3* have the ability to inhibit the growth of pathogenic fungal strains, such as fungus *C. gloeosporioides* (CD1); which causes anthracnose on fruit; fungus *F. oxysporum*, which causes wilt disease; fungus *L. theobromae N13*, which causes, stem rot disease on mango; and *N. dimidiatum* fungus, which causes brown spot disease on dragon fruit. They grow significantly over a period of 5 to 6 days. The content of the insoluble phosphate (content of P_2_O_5_) that was resolved by the three fungal strains was 341.90 1498.46, and 390.79 ppm; the content of IAA produced in the culture medium was determined to reach 49.00, 52.35, and 33.34 ppm.

*Endophytic fungi* have many important applications in agriculture, medicine, ecology, biotechnology, and other industries. They play a role in promoting plant growth and are environmentally friendly. Therefore, these fungi can be used as biopesticides and biofertilizers for plant growth and health and, thus, are capturing more attention. The results of this study can be used as a premise for further studies on the use of probiotic-containing rhizosphere fungi to re-infest soil in order to help support growth, increase resistance, and increase the yield of crops in general and medicinal plants in particular.

## Figures and Tables

**Figure 1 molecules-27-05051-f001:**
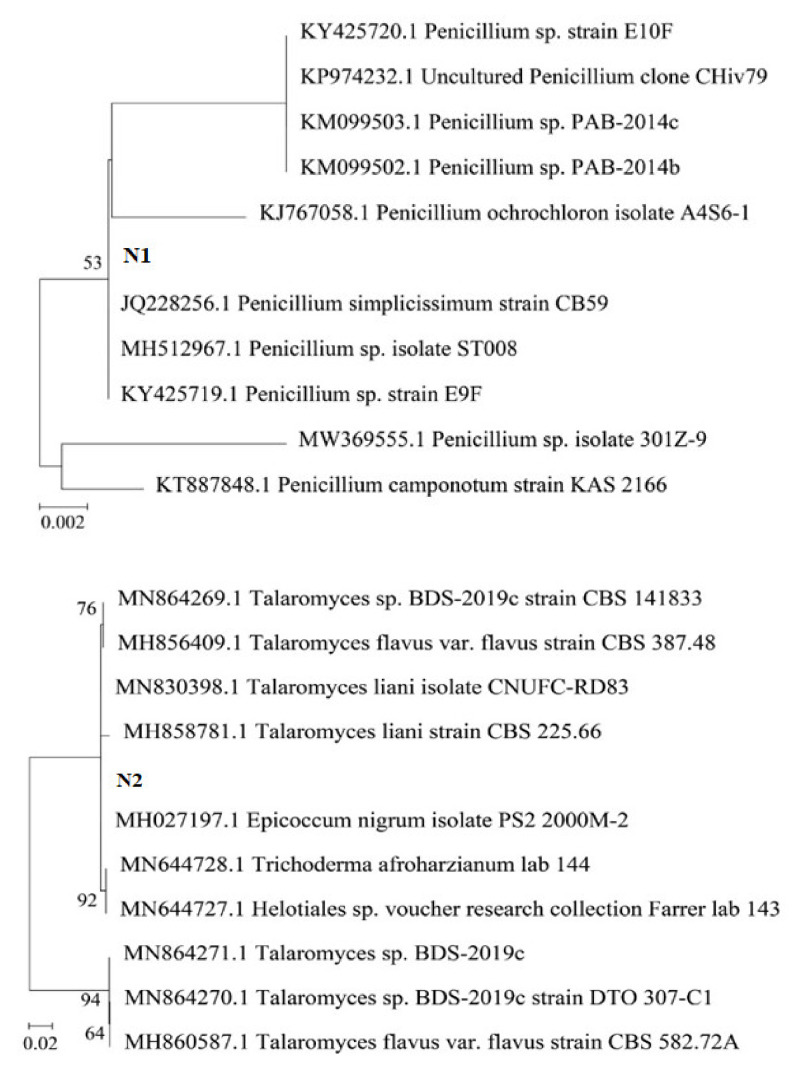
Phylogenetic tree of three fungal strains: *Penicillium simplicissimum CN7*, *Talaromyces flavus BC1*, and *Trichoderma konilangbra DL3*.

**Figure 2 molecules-27-05051-f002:**
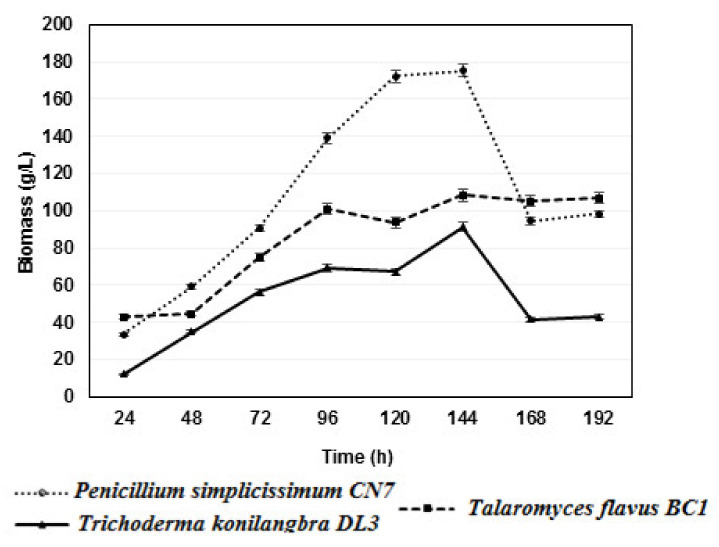
Growth dynamics of 03 rhizomatous fungal strains *Penicillium simplicissimum CN7*, *Talaromyces flavus BC1*, and *Trichoderma konilangbra DL3*.

**Figure 3 molecules-27-05051-f003:**
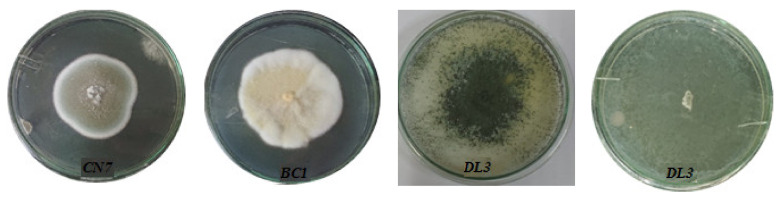
Morphology of mycelium of *Penicillium simplicissimum CN7*, *Talaromyces flavus BC1*, and *Trichoderma konilangbra DL3*.

**Figure 4 molecules-27-05051-f004:**
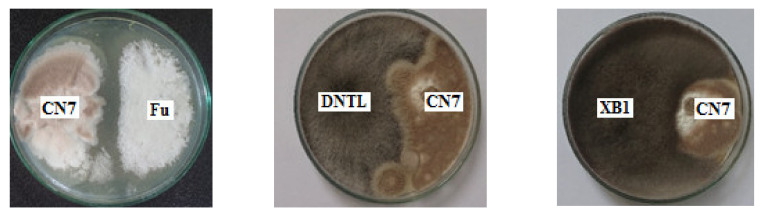
The antagonistic abilities of *Penicillium simplicissimum CN7*, *Talaromyces flavus BC1*, and *Trichoderma konilangbra DL3* with some typical plant pathogenic fungi. Note: XB1 (*L. theobromae N13*) causes stem rot in mango; CD1 (*C. gloeosporioides*) causes anthracnose disease on fruit; Fu (*F. oxysporum*) causes wilt disease; DNTL (*N. dimidiatum*) causes brown spot disease on dragon fruit.

**Table 1 molecules-27-05051-t001:** Characterization of the isolated fungal strains.

No	Mycorrhizal Plant	Strains	Colony Color	Pigment Secretion by the Phytopathogen
1	*Stevia rebaudiana (Bert.) Hemsl*	CN1	White-blue	Bright yellow color
2		CN2	Gray	None
3		CN3	White	None
4		CN4	White	None
5		CN5	Green	Purple
6		CN6	Green, clustered	Yellow
7		CN7	Light black	None
8	*Polyscias fruticosa*	DL1	Black	None
9		DL2	Black, cotton	None
10		DL3	White, cotton	None
11		DL4	Black	None
12		DL5	Yellow, cotton, clustered	Yellow
13		DL6	Black, even bacteriophage	None
14		DL7	Grizzle	None
15		DL8	White, thin yarn	None
16		DL9	White, yellow, brown thin	None
17		DL10	Light black	None
18		DL11	White short yarn	Yellow
19	*Angelica dahurica*	BC1	Gray	None
20		BC2	Dark yellow	Yellow
21		BC3	White yarn	None
22		BC4	White yarn	None
23		BC5	White yarn, crowd	None
24		BC6	White yarn, cluster, hard	None
25		BC7	White grow cluster, smooth surface	None
26		BC8	White and yellow yarn cotton	None
27		BC9	Light pink-white yarn	None

**Table 2 molecules-27-05051-t002:** Evaluation of phosphatase enzyme activity of fungal cultures.

No	Mycorrhizal Plant	Strains	Phosphatase Activities (D-d, mm)
1	*Stevia rebaudiana (Bert.) Hemsl*	CN1	0
2		CN2	0
3		CN3	7
4		CN4	0
5		CN5	14
6		CN6	0
7		CN7	12
8	*Polyscias fruticosa*	DL1	11
9		DL2	0
10		DL3	12
11		DL4	9
12		DL5	0
13		DL6	15
14		DL7	0
15		DL8	7
16		DL9	0
17		DL10	0
18		DL11	5
19	*Angelica dahurica*	BC1	10
20		BC2	0
21		BC3	0
22		BC4	11
23		BC5	0
24		BC6	14
25		BC7	11
26		BC8	12
27		BC9	7

**Table 3 molecules-27-05051-t003:** The filamentous propagation of fungal strains *Penicillium simplicissimum CN7*, *Talaromyces flavus BC1*, and *Trichoderma konilangbra DL3* on PDA agar.

Strains	*Penicillium simplicissimum CN7*	*Talaromyces flavus BC1*	*Trichoderma konilangbra DL3*
The length of filament (µm/h)	34.4 ± 2.09	43.88 ± 1.27	106.02 ± 1.60

## Data Availability

Data sharing is not applicable to this article.

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
