# Peer review of "Growth Stimulation, Phosphate Resolution, and Resistance to Fungal Pathogens of Some Endogenous Fungal Strains in the Rhizospheres of Medicinal Plants in Vietnam"

_molecules, 2022, doi:10.3390/molecules27165051_

Round 1
Reviewer 1 Report
Paper title “Study on the ability to produce growth stimulants, phosphates resolution and resistance to fungal pathogens of some endogenous fungal strains in the rhizosphere of medicinal plants in Vietnam” describes the efficacy of three endophytic fungal strains to solubilize phosphate and produce IAA. Also, the antagonistic activity against some phytopathogenic fungi was investigated. Although the challenging work was achieved by the authors, the manuscript needs major revision before being considered for publication in molecules.
1. The title is too large; it should be concise.
2. The abstract should be rephrased for varied reasons such as to be concise, should contain high promising data, and the scientific name for plants and microorganisms must be italic.
3. Please follow up the reference style according to molecules or MDPI style.
4. Line 95, “P2O5” should be “P2O5”. Please check and revise the manuscript.
5. The scientific name must be italic, please check and revise throughout the manuscript.
6. The introduction section needs to improve to highlight the activity of endophytic strains. I recommended citing of these references: https://doi.org/10.3390/plants10050935, https://doi.org/10.1007/s11816-021-00716-y,
7. Line 92 – 95, the hypothesis or aim of the work shouldn’t contain actual data, please rephrase, and add a clear hypothesis for the current study at the end of the introduction.
8. In the Material and Method section, Line 240, the authors said that they “were isolated from roots and rhizospheres of..”, then are these fungal strains, endophytes, or rhizosphere strains? please use the appropriate word.
9. Authors must refer to the source of each fungal isolate to avoid confusion, such as N1 from roots of Stevia rebaudiana (Bert.), or Polyscias fruticosa, or Angelica dahurica and so on,
10. In this section, the authors should mention the number of replicates from each plant, the number of fungal isolates from each sample, the number of each part in plates during isolation, and the number of replicates during isolation. All this data should be discussed in more detail.
11. How did the authors confirm these strains are endophytes and not epiphytes or rhizosphere strains?
12. The surface sterilization and isolation procedures should be discussed in more detail, please see the following references: https://doi.org/10.3390/biom11020140;
13. Line 246; “PVK medium” please mentioned the abbreviation complete the first time. Please check and revise the manuscript.
14. Line 250, “photometric method” this method should be mentioned in more detail.
15. Lines 256 and 257, these components for 100 mL or one liter, and why select 1 g of tryptophan?
16. In lines 273 and 247, the authors should mention the source of these fungal strains.
17. Results and Discussion section, the authors should be clarifying the total number of fungi that were isolated from the roots of three selected plants Stevia rebaudiana (Bert.), or Polyscias fruticosa, or Angelica dahurica, and refer to the experiment (s) with their data that used to select the three most potent fungal strain N1, N2, and N3 followed by identification of the most potent fungal isolates.
18. Line 108 and 109, this title can be summarized as “Identification of most potent endophytic fungal strains”.
19. Line 110 – 115, this paragraph can be transferred to the material and method section.
20. Please add the GenBank accession number of these three most potent endophytic fungal strains.
21. Line 120 – 130, the authors mentioned some fungal strains from published studies that have been recognized as phosphate solubilization and IAA producers and said the obtained fungal strains belonging to these strains and hence can be solubilizing the phosphate and producing IAA without adding the actual data of these experiments. This data is considered the main backbone of the current study and hence it must be added to this data.
22. Line 142, the authors mentioned that the highest fungal biomass was obtained at “at 37oC”, please check this value, because the optimum temperature for fungal growth is in the range of 25 – 30 °C.
23. In Figure 2, the biomass (g/L) of fungal strains N1 and N3 reached the highest at 144 h and decreased after that, please clarify how biomass decreased after that, in my opinion, it should be stable and not decrease with time as in N2.
24. The scientific name in legend and title of figure 2, 3, 4, and Table 1 must be italic.
25. What is the difference between titles 2.2 and 2,3.
26. The quality of Figure 4 is high poor, please add a clearer photo.
27. The data in the current study need deep discussion.
28. The manuscript contains high typo-error and grammatical mistakes, please revised it carefully.
Author Response
Dear Reviewer
We report the response to each of the points raised by the referees and a summary of the changes made. We added our response and the list of changes made for each of the referees’ comments.
We included a revised version of the manuscript in which all the changes made are highlighted in yellow as “Revised Manuscript”.
We think that the paper was significantly improved following the referees’ advice. We hope that our answers and the changes made can be considered satisfactory.
With best regards,
On behalf of the authors
Vu Duc Chinh, Nguyen Thi Mai Huong and Pham Thi Thu Hoai.

Reviewer 2 Report
Review for
Study on the ability to produce growth stimulants, phosphates resolution and resistance to fungal pathogens of some endogenous fungal strains in the rhizosphere of medicinal plants in Vietnam
In this paper, three strains of endophytic fungi N1, N2, and N3 were isolated from the roots of Stevia rebaudiana (Bert.) Hemsl., Polyscias fruticosa and Angelica dahurica in some localities such as Hung Yen, Phu Tho, Lao Cai, Nam Dinh provinces in Vietnam.
Vietnam is a strong player in the world medicinal plant industry
A Review of Traditional Uses, Phytochemistry and Pharmacological Properties of Some Vietnamese Wound-Healing Medicinal Plants
Hua, O.H., Tran, Q.T.T., Trinh, D.-T.T., (...), Duong, D.P.N., Nguyen, T.T. 2022 Natural Product Communications 17(4)
+ interest in Endophytic Microorganisms
Notes on Culturable Endophytic Microorganisms Isolated from 14 Medicinal Plants in Vietnam: A Diversity Analysis to Predict the Host-Microbe Correlations Tran, H.M., Nguyen, D.T.T., Mai, N.T., (...), Nguyen, H.Q., Pham, H.T. 2022 Current Microbiology 79(5),140
Base on some morphological characteristics and internal transcribed spacer gene sequence analysis of fungal strains N1, N2, N3 were named Penicillium sp. N1, Talaromyces sp. N2 and Trichoderma sp. N3.
Authors should provide species level identification, through multi gene sequencing/phylogeny
Writing and English are not satisfactory.
They grow significantly at the time from 5 to 6 days.
Such statement could not be written in a scientific paper.
Very preliminary study.
Endophytic fungi strains Penicillium sp. N1, Talaromyces sp. N2 and Trichoderma sp. N3 295 have the ability to inhibit the growth of pathogenic fungal strains such as fungus C. gloe-296 osporioides (CD1) causing anthracnose on fruit, fungus F. oxysporum causing wilt disease, 297 fungus L. theobromae N13 causing stem rot disease on mango and N. dimidiatum fungus 298 causing brown spot disease on dragon fruit. They grow significantly at the time from 5 to 299 6 days.
Why do you target such pathogenic fungal strains (stem rot disease on mango, brown spot disease on dragon fruit)?
Not pathogenic fungal strains involved in medicinal plant diseases?
Previous patents or industrial applications with Endophytic fungi strains (coming from medicinal plants) applied to plant protection?
Author Response

(The authors gave the same response as above.)

Round 2
Reviewer 1 Report
Although the authors perform significant efforts to improve the manuscript and answer some issues, various serious issues still needed to be clarified by authors.
1- Again, based on lines 138 – 141, the fungal strains N1, N2, and N3 are endophytic strains or rhizospheric strains. Please clarify
2- The authors did not mention their actual data (phosphate solubilizing and IAA) and only referred to it. These data are considered the main results of the current study. I recommend adding a table containing the data of 27 fungal isolates and highlighting the most potent data.
3- Line 163 – 252. The authors increase the discussion of phosphate solubilizing activity and mention data such as field experiments and greenhouse experiments not related to the current study, especially authors who do not perform field or greenhouse experiments. Please concise this part.
4- In the ligand of figure 2, the fungal names must be italic.
5- Line 376 and 377, what is the meaning of “light absorbed in the ultraviolet (UV) to visible (VIS) to infra-red (IR) range”, please write the method in more detail.
6- The authors did not mention the isolation procedures, number of replicates from each plant, the number of fungal isolates from each sample, the number of each part in plates during isolation, and the number of replicates during isolation. All these data should be discussed in more detail in the material and method section.
7- In the author's response to the comment “The surface sterilization and isolation procedures should be discussed in more detail,”, the authors mentioned this data was written in the first 6 lines in section 2.1 and this is not correct. This data should be added in the material and method section under the title “isolation of endophytic fungal strains”.
8- Line 384, “0.1g tryptophan” 0.1 g per what????, (mL or 100 mL or 1 L)??, and why select this concentration, please clarify (Please take attention to this comment from round 1).
9- The authors in their response mentioned that the “Source of these fungal strains was mentioned in Item 3.1. Materials.” and this is not correct, please add the source of phytopathogenic fungi.
10- The accession numbers that were added by the authors for the fungal strains were OL795992, OL795993, and OL875113 for the fungal strain Penicillium sp. N1, Talaromyces sp. N2, and Trichoderma sp. N3. By searching on GenBank, I found these accession numbers for Penicillium simplicissimum strain CN7, Talaromyces flavus isolate BC1, and Trichoderma konilangbra strain DL3. Why do authors give the current strain the name of genus only with a different code than deposited in GenBank?
11- Please take attention to cite the reference in the correct manner (authors name), for instance, references 1, 2, 12, 13, ….
12- Line 141, “Felde et al. (2006) [22]” should be “Felde et al. [22]” please revise throughout the manuscript.
13- Line 147, “T. flavus” please write it complete the first time.
Author Response
Although the authors perform significant efforts to improve the manuscript and answer some issues, various serious issues still needed to be clarified by authors.
1-Again, based on lines 138 – 141, the fungal strains N1, N2, and N3 are endophytic strains or rhizospheric strains. Please clarify
Authors’ reply: The fungal strains N1, N2, and N3 are endophytic strains.
2-The authors did not mention their actual data (phosphate solubilizing and IAA) and only referred to it. These data are considered the main results of the current study. I recommend adding a table containing the data of 27 fungal isolates and highlighting the most potent data.
Authors’ reply: We thank the reviewer for the suggestion. Table 1 and 2 containing the data of 27 fungi isolated were completed in item 2.1.
3-Line 163 – 252. The authors increase the discussion of phosphate solubilizing activity and mention data such as field experiments and greenhouse experiments not related to the current study, especially authors who do not perform field or greenhouse experiments. Please concise this part.
Authors’ reply: Thank you for this suggestion. The field experiments and greenhouse experiments not related to the current study were removed.
4-In the ligand of figure 2, the fungal names must be italic.
Authors’ reply: The fungal names in figure 2 changed to be italic.
5-Line 376 and 377, what is the meaning of “light absorbed in the ultraviolet (UV) to visible (VIS) to infra-red (IR) range”, please write the method in more detail.
Authors’ reply: The method was written in more detail in item 3.2.2
6-The authors did not mention the isolation procedures, number of replicates from each plant, the number of fungal isolates from each sample, the number of each part in plates during isolation, and the number of replicates during isolation. All these data should be discussed in more detail in the material and method section.
Authors’ reply: The isolation procedures were completed in 3.2.1. In item 2.1, paragraph 2, table 1 and 2 mentioned the issues above.
7-In the author's response to the comment “The surface sterilization and isolation procedures should be discussed in more detail,”, the authors mentioned this data was written in the first 6 lines in section 2.1 and this is not correct. This data should be added in the material and method section under the title “isolation of endophytic fungal strains”.
Authors’ reply: We thank the reviewer for the suggestion. It was added in the material and method section under the title 3.2.1 “Isolation of endophytic fungal strains”
8-Line 384, “0.1g tryptophan” 0.1 g per what????, (mL or 100 mL or 1 L)??, and why select this concentration, please clarify (Please take attention to this comment from round 1).
Authors’ reply: The concentration of tryptophan is 0.1g/100mL according to O.A. Apine and J.P. Jadhav, Journal of Applied Microbiology 110, 1235–1244 (2011).
9-The authors in their response mentioned that the “Source of these fungal strains was mentioned in Item 3.1. Materials.” and this is not correct, please add the source of phytopathogenic fungi.
Authors’ reply: It was added.
10-The accession numbers that were added by the authors for the fungal strains were OL795992, OL795993, and OL875113 for the fungal strain Penicillium sp. N1, Talaromyces sp. N2, and Trichoderma sp. N3. By searching on GenBank, I found these accession numbers for Penicillium simplicissimum strain CN7, Talaromyces flavus isolate BC1, and Trichoderma konilangbra strain DL3. Why do authors give the current strain the name of genus only with a different code than deposited in GenBank?
Authors’ reply: Thank a lot for this suggestion. We changed their names throughout the manuscript.
11-Please take attention to cite the reference in the correct manner (authors name), for instance, references 1, 2, 12, 13, ….
12-Line 141, “Felde et al. (2006) [22]” should be “Felde et al. [22]” please revise throughout the manuscript.
Authors’ reply: We thank the reviewer for the suggestion. We revised throughout the manuscript.
13-Line 147, “T. flavus” please write it complete the first time.
Authors’ reply: It was revised.

Reviewer 2 Report
The authors have addressed my comments and problems seriously. The revised version is more comprehensive and it takes into account the reviewers suggestions. Therefore I recommend its acceptance in the revised form.
Author Response
The authors have addressed my comments and problems seriously. The revised version is more comprehensive and it takes into account the reviewers suggestions. Therefore I recommend its acceptance in the revised form.
Authors’ reply: We thank a lot the reviewer for your support.
